# MMM-RS: A Multi-modal, Multi-GSD, Multi-scene Remote Sensing Dataset and Benchmark for Text-to-Image Generation

**Jialin Luo[1,*], Yuanzhi Wang[1,*], Ziqi Gu[1], Yide Qiu[1], Shuaizhen Yao[1], Fuyun Wang[1], Chunyan Xu[1], Wenhua Zhang[1], Dan Wang[2], Zhen Cui[1,†]**

1. PCA Lab, Key Lab of Intelligent Perception and Systems for High-Dimensional Information of Ministry of Education, School of Computer Science and Engineering, Nanjing University of Science and Technology, Nanjing, China.
2. Beijing Institute of Spacecraft System Engineering, Beijing, China.

## Abstract

Recently, the diffusion-based generative paradigm has achieved impressive general image generation capabilities with text prompts due to its accurate distribution modeling and stable training process. However, generating diverse remote sensing (RS) images that are tremendously different from general images in terms of scale and perspective remains a formidable challenge due to the lack of a comprehensive remote sensing image generation dataset with various modalities, ground sample distances (GSD), and scenes. In this paper, we propose a *Multi-modal, Multi-GSD, Multi-scene Remote Sensing (**MMM-RS**)* dataset and benchmark for text-to-image generation in diverse remote sensing scenarios. Specifically, we first collect nine publicly available RS datasets and conduct standardization for all samples. To bridge RS images to textual semantic information, we utilize a large-scale pretrained vision-language model to automatically output text prompts and perform hand-crafted rectification, resulting in information-rich text-image pairs (including multi-modal images). In particular, we design some methods to obtain the images with different GSD and various environments (e.g., low-light, foggy) in a single sample. With extensive manual screening and refining annotations, we ultimately obtain a MMM-RS dataset that comprises approximately 2.1 million text-image pairs. Extensive experimental results verify that our proposed MMM-RS dataset allows off-the-shelf diffusion models to generate diverse RS images across various modalities, scenes, weather conditions, and GSD. The dataset is available at https://github.com/ljl5261/MMM-RS.

## 1 Introduction

Remote sensing (RS) image, as a domain-specific image, plays an important role in the applications of sustainable development for human society, such as disaster response, environmental monitoring, crop yield estimation, and urban planning [29] [1] [38] [36]. Over the last decade, RS-based deep learning models have demonstrated substantial success in various computer vision tasks such as scene classification [8], object detection [41], semantic segmentation [18], and change detection [15], which can facilitate the above applications of sustainable development. Nevertheless, these models may suffer from limited performance due to the lack of large-scale high-quality dataset, and obtaining RS images is often not easy and expensive (i.e., need to launch a RS satellite).

---

[*]Co-first Authors, Equal Contribution: Jialin Luo, Yuanzhi Wang
[†]Corresponding Author: Zhen Cui

38th Conference on Neural Information Processing Systems (NeurIPS 2024) Track on Datasets and Benchmarks.

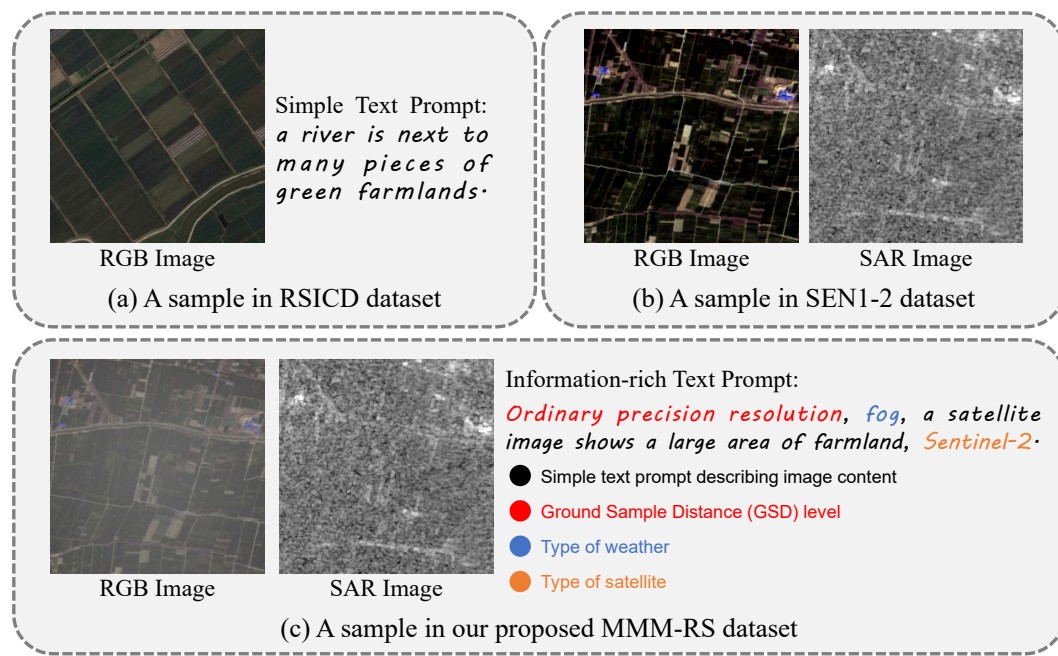

Figure 1: The samples from different RS dataset. (a) shows a sample in the classic RSICD dataset [17], which is a simple text-image pair. (b) shows a sample in the classic SEN1-2 dataset [25], which is a multi-modal image pair including RGB image and SAR image. (c) shows a sample in our proposed MMM-RS dataset. In contrast, our MMM-RS dataset provides not only multi-modal image pair but also information-rich text prompt including simple text prompt describing image content, GSD level (i.e., spatial resolution), type of weather and satellite (different color for better observation).

To address the above issues, a straightforward way is to utilize the existing datasets and advanced generative models [23, 33, 32, 34] to train a RS-based text-to-image generation model, and then the user can obtain the diverse RS images via inputting text prompts. However, there exist some challenges that may cause the trained model to fail to satisfy the user's requirements. As shown in Fig. 1 (a), we show a sample in the classic RS dataset RSICD [17], which is a text-image pair with the simple text description. Intuitively, the RSICD dataset does not allow models to generate multi-modal RS images. In Fig. 1 (b), we show a sample from another classic dataset SEN1-2 [25], which is a multi-modal RS image pair (including RGB image and Synthetic Aperture Radar (SAR) image) but does not include text descriptions. Thus, the SEN1-2 dataset cannot allow models to generate RS images from text prompts. From the phenomenon described above, we can observe that there is no publicly available RS dataset that contains both multi-modal RS images and information-rich text descriptions for diverse and comprehensive RS image generation. In addition, we survey the currently mainstream RS datasets and report their key properties statistically in Tab. 1, which further provides evidence for the above observation.

To this end, we propose a Multi-modal, Multi-GSD, Multi-scene Remote Sensing (MMM-RS) dataset and benchmark for text-to-image generation in diverse remote sensing scenarios. We first collect 9 publicly available RS datasets and standardize all samples to a uniform size. Then, to inject the textual semantic information in each sample for conducting text-to-image generation, we utilize a large-scale pretrained vision-language model, i.e., BLIP-2 [13] to automatically output text prompt describing each RS image context. To provide the various ground sample distances (GSD) samples, we design a GSD sample extraction strategy to extract different GSD images for each sample and define the GSD-related text prompts describing different GSD levels. In particular, we select some RGB samples to be used for synthesizing samples of different scenes (e.g., snowy, fog), thus empowering the well-trained model with the ability to perceive the various scenes. Finally, with extensive manual screening and refining annotations (i.e., text prompts), we obtain approximately 2.1 million well-crafted and information-rich text-image pairs to result in our MMM-RS dataset. As shown in Fig. 1 (c), we show a sample in our MMM-RS dataset. In contrast, the MMM-RS dataset provides not only a multi-modal image pair but also an information-rich text prompt. With our MMM-RS dataset, we can train a

Table 1: Comparisons of mainstream RS datasets. "simple" denotes the simple text description (e.g., Fig. 1 (a)). "information-rich" denotes the information-rich text description (e.g., Fig. 1 (c)). "Multi-modal" means remote sensing imaging content captured by different sensors, such as RGB image, Synthetic Aperture Radar (SAR) image, and Near Infrared (NIR) image.

| Dataset | Text Descriptions | Multi-modal | GSD Descriptions | Multi-GSD | Weather |
|---------|-------------------|-------------|------------------|-----------|---------|
| MRSSC2.0 | × | × | × | × | × |
| Inria | × | × | × | × | × |
| NaSC-TG2 | × | × | × | × | × |
| HRSC2016 | × | × | × | × | × |
| TGRS-HRRSD | × | × | × | × | × |
| fMoW | × | × | × | × | × |
| SAMRS | × | × | × | × | × |
| GID | × | ✓ (RGB, NIR) | × | × | × |
| DDHRNet | × | ✓ (RGB, SAR) | × | × | × |
| WHU-OPT-SAR | × | ✓ (RGB, SAR, NIR) | × | × | × |
| SEN1-2 | × | ✓ (RGB, SAR) | × | × | × |
| RSICD | ✓ (simple) | × | × | × | × |
| UCM-Captions | ✓ (simple) | × | × | × | × |
| NUPM-Captions | ✓ (simple) | × | × | × | × |
| RS5M | ✓ (simple) | × | ✓ | × | × |
| SkyScript | ✓ (simple) | × | × | ✓ | × |
| MMM-RS (Ours) | ✓ (information-rich) | ✓ (RGB, SAR, NIR) | ✓ | ✓ | ✓ |

RS text-to-image generation model by fine-tuning the off-the-shelf text-to-image diffusion models (e.g., Stable Diffusion [23], ControlNet [39]) for generating multi-modal, multi-GSD, multi-scene RS images. In summary, the contributions of this work can be concluded as:

- We construct a large-scale Multi-modal, Multi-GSD, and Multi-scene Remote Sensing (MMM-RS) dataset and benchmark for text-to-image generation in diverse RS scenarios, which standardizes 9 publicly available RS datasets with uniform and information-rich text prompts.

- To provide the various GSD samples, we design a GSD sample extraction strategy that extracts different GSD levels images for each sample and define the GSD-related text prompts describing different GSD levels. Furthermore, due to the lack of real-world multi-scene samples, we select some RGB samples and utilize existing techniques to synthesize samples with different scenes including fog, snow, and low-light environments.

- We use our proposed MMM-RS dataset to fine-tune the advanced Stable Diffusion, and perform extensive quantitative and qualitative comparisons to prove the effectiveness of our MMM-RS dataset. In particular, we use the aligned multi-modal samples (including RGB, SAR, and infrared modalities) in the MMM-RS dataset to train the cross-modal generation models based on ControlNet, and the visualization results demonstrates impressive cross-modal generation capabilities.

## 2  Background

Remote sensing (RS) imaging data are widely used in various computer vision tasks such as scene classification [37, 3, 45, 16, 30], object detection [40, 35, 12], segmentation [28, 14, 26, 30], change detection [2, 27, 7], and RS image caption [22, 17, 4, 10, 43]. For scene classification, the classic UC Merced Land Use Dataset [37] contains 21 scene classes, and each class has 100 images. MRSSC2.0 [16] is a multi-modal remote sensing scene classification dataset that contains 26,710 images of 7 typical scenes such as city, farmland, mountain, etc. In object detection field, the classic dataset HRSC2016 [40] consists of 1,070 images with 2,976 ship bounding boxes for ship detection in RS scenarios. For segmentation task, GID [28] contains 150 large-size ($7200 \times 6800$) RS image with fine-grained pixel-level annotations. WHU-OPT-SAR [14] is a multi-modal segmentation dataset containing three diverse modalities, i.e., RGB, SAR, and NIR. For change detection task, the LEVIR-CD [2], Hi-UCD [27], and CDD [7] are used to train a model predicting the changes in the same region. In the RS image caption domain, the classic datasets, such as UCM-Captions [22], RSICD [17], NWPU-Captions [4], contain images with simple text descriptions to conduct image-to-text transferring. Despite the great success, there is no publicly available dataset for RS text-to-image generation task. In this work, we aim to propose a Multi-modal, Multi-GSD, Multi-scene RS dataset and benchmark for text-to-image generation in diverse RS scenarios.

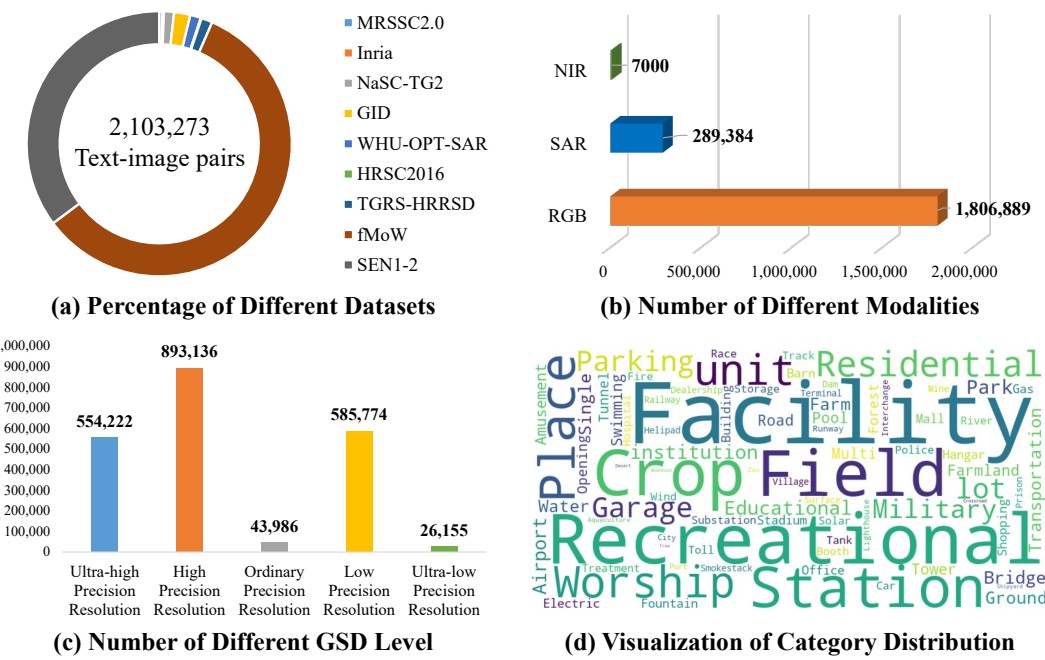

**(a) Percentage of Different Datasets**

**(b) Number of Different Modalities**

**(c) Number of Different GSD Level**

**(d) Visualization of Category Distribution**

Figure 2: MMM-RS dataset statistics from different aspects.

## 3 MMM-RS Dataset

### 3.1 Dataset Statistics

This section provides basic statistics of the MMM-RS dataset. The MMM-RS dataset is derived from 9 publicly available RS datasets: MRSSC2.0 [16], Inria [19], NaSC-TG2 [45], GID [28], WHU-OPT-SAR [14], HRSC2016 [40], TGRS-HRRSD [42], fMoW [5], and SEN1-2 [25]. With standardized processing, MMM-RS finally contains 2,103,273 text-image pairs, and the percentage of different datasets is presented in Fig. 2 (a).

**Statistics for Different Modalities.** The MMM-RS dataset contains three modalities: RGB image, Synthetic Aperture Radar (SAR) image, and Near Infrared (NIR) image. Note that the three modalities are aligned. Fig. 2 (b) shows the number of different modalities, we can observe that the number of RGB modality is 1,806,889 which dominates the entire dataset. In contrast, the number of SAR modality and NIR modality are much smaller, with the number of 289,384 and 7000, respectively. This is because the three modal-aligned samples are very difficult to obtain requiring multiple simultaneous satellites to perform computational imaging of the same region [14].

**Statistics for Different Ground Sample Distance (GSD) Levels.** Fig. 2 (c) shows the number of different GSD levels, in which the all samples are standardized five category: Ultra-high Precision Resolution (GSD < 0.5 m/pixel), High Precision Resolution (0.5 m/pixel ≤ GSD < 1 m/pixel), Ordinary Precision Resolution (1 m/pixel ≤ GSD < 5 m/pixel), Low Precision Resolution (5 m/pixel ≤ GSD < 10 m/pixel), and Ultra-low Precision Resolution (GSD ≥ 10 m/pixel).

**Visualization of Category Distribution.** Fig. 2 (d) visualizes the category distribution of MMM-RS, we can observe that the categories are mainly focused on *"Recreational Facility"* and *"Crop Field"*. The reason behind this phenomenon is that more than half of the samples in the MMM-RS are from fMoW [5] that is dominated by two categories *"Recreational Facility"* and *"Crop Field"*.

### 3.2 Dataset Preprocessing and Standardization

The sizes of the samples in different datasets are often inconsistent, thus the first step in the dataset construction is to standardize all samples. Referring to the most popular open source text-to-image diffusion model, i.e., Stable Diffusion [23], all samples are standardized to a uniform size of $512 \times 512$. The algorithm for dataset standardization is illustrated in Algorithm 1. Concretely, for samples with

---
**Algorithm 1** Dataset Standardization
---
**Input:** Original dataset $\mathcal{X}$, ESRGAN$(\cdot)$ with the scale factor $\times 2$, image size extractor Size$(\cdot)$
**Output:** Standardized dataset $\widetilde{\mathcal{X}}$
  **for** $X$ in $\mathcal{X}$ **do**
    **if** Size$(X) > (512, 512)$ **then**
      Crop all non-overlapping images $\{\widetilde{X}_1, \widetilde{X}_2, \cdots, \widetilde{X}_n\}$ with the image size of $(512, 512)$;
      $\{\widetilde{X}_1, \widetilde{X}_2, \cdots, \widetilde{X}_n\}$ is appended to $\widetilde{\mathcal{X}}$;
    **else**
      Calculate the minimum $L$ of the height and the width: $L = \min(\text{Size}(X))$;
      Crop a image $X_{\text{crop}}$ with the size of $(L, L)$;
      $X_{\text{crop}}$ is super-resolved to $X_{\text{SR}} = \text{ESRGAN}(X_{\text{crop}})$ with the size of $(2L, 2L)$;
      Resize $X_{\text{SR}}$ to $\widetilde{X}$ with the size of $(512, 512)$ by Bicubic Interpolation;
      Calculate the GSD $G_{\widetilde{X}}$ of $\widetilde{X}$ according to the GSD $G_X$ of $X$: $G_{\widetilde{X}} = \frac{L}{512} \times G_X$;
      $\widetilde{X}$ is appended to $\widetilde{\mathcal{X}}$;
    **end if**
  **end for**
---

the size higher than $512 \times 512$, we crop all non-overlapping images with size of $512 \times 512$ as new samples. For samples with the size lower than $512 \times 512$, we first calculate the minimum $L$ of the height and the width, and then crop a image with the size of $L \times L$. Note that the sizes of the samples in the original dataset are greater than or equal to $256 \times 256$. To preserve more high-frequency detail while upsampling the images, an ESRGAN [31] super-resolution model with the scale factor $\times 2$ is introduced (denoted as ESRGAN$(\cdot)$) to super-resolve the cropped image. Finally, we use Bicubic interpolation to resize the super-resolved image and output the standardized image with the size of $512 \times 512$. In addition, we need to update the GSD of sample according to the change in image size.

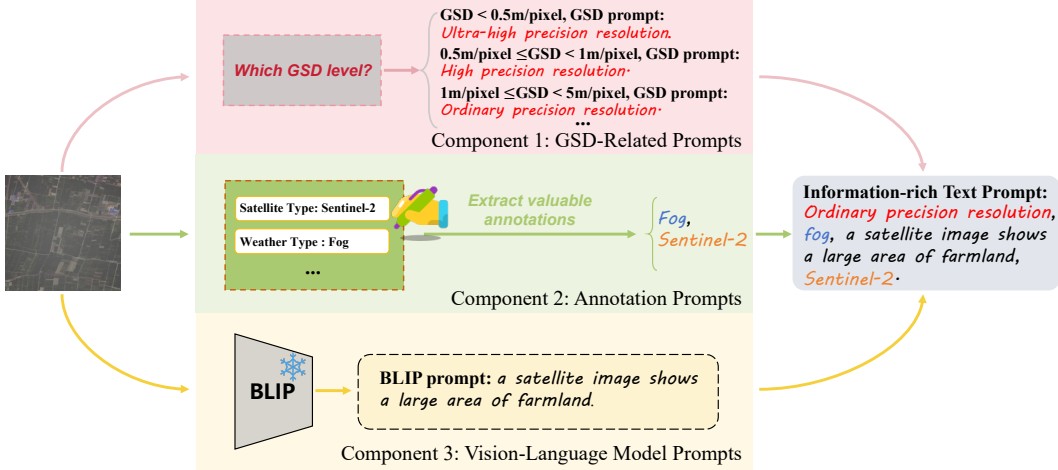

Figure 3: The framework of information-rich text prompt generation.

## 3.3 Information-rich Text Prompt Generation

We now elaborate on how to generate an information-rich text prompt for each sample. The framework is shown in Fig. 3, which consists of three components: GSD-related prompts, annotation prompts, and vision-language model prompts. For the GSD-related prompts, we output the GSD prompt of input image according to the predefined GSD level in Sec. 3.1. For the annotation prompts, we first extract the annotation contents such as satellite type, weather type, category, etc. that may (or may not, e.g., SEN1-2 [25] does not include annotations) exist in the original datasets, and then the satellite type and weather type are extracted as output. The category information is used for final manual text prompt proofreading. For the vision-language model prompts, we aim to utilize the pretrained large-scale vision-language model BLIP-2 [13] to output a simple text prompt describing input image content. Finally, we combine the outputs of the above three components and obtain

an information-rich text prompt. Furthermore, we exploit the category information and conduct extensive manual screening and refining to improve the accuracy of the information-rich text prompts.

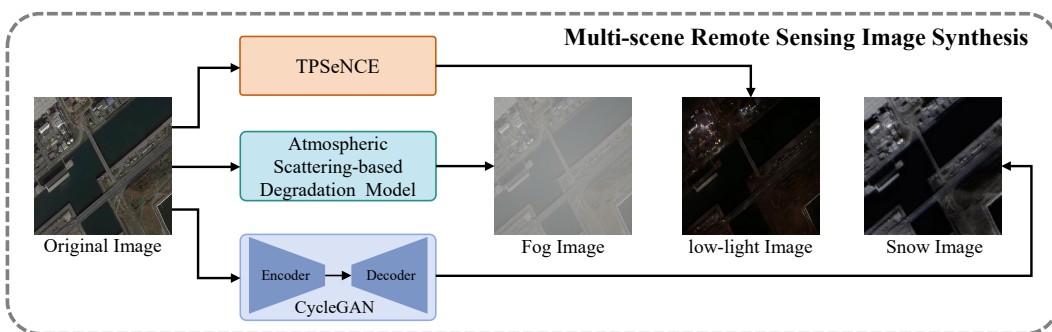

Figure 4: The framework of multi-scene remote sensing image synthesis.

## 3.4 Multi-scene Remote Sensing Image Synthesis

To address the problem of multi-scenes RS data scarcity, we aim to synthesize some common scene data by leveraging existing techniques. Specially, we select 10,000 samples from standardized dataset to be used for synthesizing images with three common scenes: fog scene, snow scene, and low-light scene. The overview framework of multi-scene RS image synthesis is shown in Fig. 4.

**Fog scene synthesis.** To synthesize fog image, we use the classic atmospheric scattering-based degradation model [6] to generate photorealistic fog images.

**Low-light scene synthesis.** For synthesizing RS images in the low-light scene, we leverage the latest low-light image generation model TPSeNCE [44] to synthesize realistic low-light RS images. In practice, we directly use the pretrained model provided by the authors to generate low-light RS images because it is sufficient to fit the RS scenarios.

**Snow scene synthesis.** For synthesizing snow RS images, a straightforward way is to use TPSeNCE model to generate target images, as TPSeNCE also provides the pretrained snow image generation model. However, in practice, we observe that the pretrained model is difficult to synthesize snow RS images. To address the above issue, we first screen all RS images containing snow scene in the dataset (460 images in total) and select another 460 RS images that do not contain snow scene. Then, we utilize the CycleGAN [46] that is an unpaired image-to-image translation model and use the above selected unpaired data to train a clear-to-snow generation model based on CycleGAN. Finally, we use the well-trained model to synthesize the snow RS images.



Figure 5: An example for generating different GSD images for the same sample.

## 3.5 Generating Different GSD Images for the Same Sample.

Existing RS datasets often contain various GSD image, e.g., the fMoW [5] contains images with GSD ranging from 0.5 m/pixel to 2 m/pixel, the GSD of the SEN1-2 [25] is 10 m/pixel, and the GSD of the MRSSC2.0 [16] is 100 m/pixel. However, there is no dataset that contains different GSD images for a single sample. In other words, these datasets cannot allow the model to generate images with different GSDs for the same scene. To address the above issues, we design a GSD sample

extraction strategy to extract different GSD images for each sample. The main idea of this strategy is to crop images with different sizes (same height and width) from a large-size RS image and ensure that the cropped images of different sizes have obvious GSD changes. Then, all cropped images are standardized to the size of $512 \times 512$, so that the GSD of standardized images can be computed as $G_{std} = (L/512) \times G_{ori}$, where $G_{std}$ and $G_{ori}$ denote the GSD of the standardized image and original image, respectively. $L$ denotes the height and width of the cropped image.

In practice, we perform the above strategy on the Inria dataset [19] to generate different GSD images because its image size is $5000 \times 5000$ that is enough to crop images with different sizes. As shown in Fig. 5, we show an example of generating different GSD images for the same sample. Specially, the size and GSD of the original image are $5000 \times 5000$ and 0.3 m/pixel, respectively. We then crop four images with four different sizes (i.e., $4096 \times 4096$, $2048 \times 2048$, $1024 \times 1024$, and $512 \times 512$) from the original image. Note that the higher resolution images completely cover the lower resolution images, which ensures that all cropped images maintain consistent scene content. Finally, with the four cropped images, we standardize them to a uniform size of $512 \times 512$, and the GSD is updated to 2.4 m/pixel, 1.2 m/pixel, 0.6 m/pixel, and 0.3 m/pixel, respectively. The above process could facilitate the generation of various GSD images, ensuring that the model can perceive variations between different GSDs while maintaining scene consistency.

## 4 Experiments

### 4.1 Fine-tuning Stable Diffusion for RS Text-to-Image Generation

To validate the effectiveness of the MMM-RS dataset in the RS text-to-image generation task, we use this dataset along with the currently prominent Stable Diffusion [23] to achieve RS Text-to-Image Generation. Those experiments are a crucial part of our work.

**Experiment settings.** Our experiments utilize the Stable Diffusion-V1.5 model [23] (called by SD1.5) as the foundational pre-trained model. To optimize its performance for our specific requirements in RS text-to-image generation, the LoRA [11] technique is adopted to update the stable diffusion model. In the generative phase, our generative model undergoes a training regimen of 200,000 iterations on our MMM-RS datasets. We use a learning rate of 0.0001 and employ the Adam optimizer to ensure effective training. Our text prompt, which is crucial for directing the image generation process, is meticulously constructed using a combination of four components: {Ground Sample Distance level}, {Type of weather}, {Simple text prompt describing image content}, and {Type of satellite} (such as: *High precision resolution, snow, a satellite image shows a park in the city, Google Earth*). This method allows us to explore various textual inputs and their impact on the generated images. To evaluate our generative model's performance, we utilize two widely recognized metrics: the Frechet Inception Distance [9] (FID) and the Inception Score [24] (IS). These metrics are crucial for assessing the quality and diversity of the images generated by our model, allowing us to compare them against real images in terms of their distribution and visual clarity. We conduct all experiments using the PyTorch framework on 8 NVIDIA RTX 4090 GPUs.

Table 2: Quantitative comparison with evaluated baselines.

| Metrics | SD1.5 [23] | SD2.0 [23] | SDXL [21] | DALL-E 3 [20] | **Ours** |
|---|---|---|---|---|---|
| FID $\downarrow$ | 172.78 | 175.68 | 168.34 | 347.88 | **92.33** |
| IS $\uparrow$ | 6.64 | 6.31 | 6.89 | 2.63 | **7.21** |

**Quantitative comparisons.** To demonstrate the effectiveness of the MMM-RS dataset in the RS text-to-image generation task, we conduct the quantitative comparisons in terms of the FID and the IS metrics across different generative models, as shown in Tab. 2. It should be noted that to ensure a fair comparison of model performance, each model generated 500 RS images for the calculation of the above metrics. Notably, our model achieves a lower FID value compared to other models. A lower FID indicates that the images generated by our model have a closer statistical distribution to real images, suggesting higher quality and accuracy of the generated images. Additionally, our model scores the highest on the IS, surpassing the other generative models. The higher IS indicates that our model not only produces more diverse images but also maintains better clarity and recognizability in the generated images. These results underscore the effectiveness of the MMM-RS dataset in enhancing the capabilities of generative models for RS image generation. The substantial improvements in both

metrics compared to existing models highlight the potential of our tailored approach in producing realistic and varied images, suitable for advanced remote sensing applications. This validates the utility of the MMM-RS dataset as a valuable resource in the field of generative image modeling.

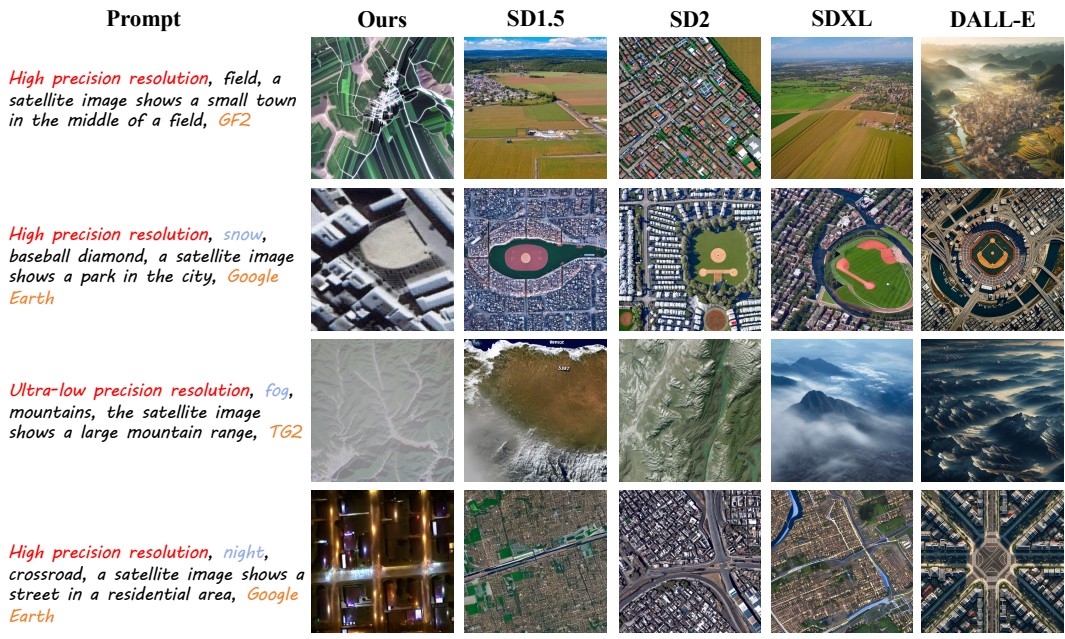

| Prompt | Ours | SD1.5 | SD2 | SDXL | DALL-E |

*High precision resolution, field, a satellite image shows a small town in the middle of a field, GF2*

*High precision resolution, snow, baseball diamond, a satellite image shows a park in the city, Google Earth*

*Ultra-low precision resolution, fog, mountains, the satellite image shows a large mountain range, TG2*

*High precision resolution, night, crossroad, a satellite image shows a street in a residential area, Google Earth*

Figure 6: Visualization results of different methods in multi-scene RS image generation.

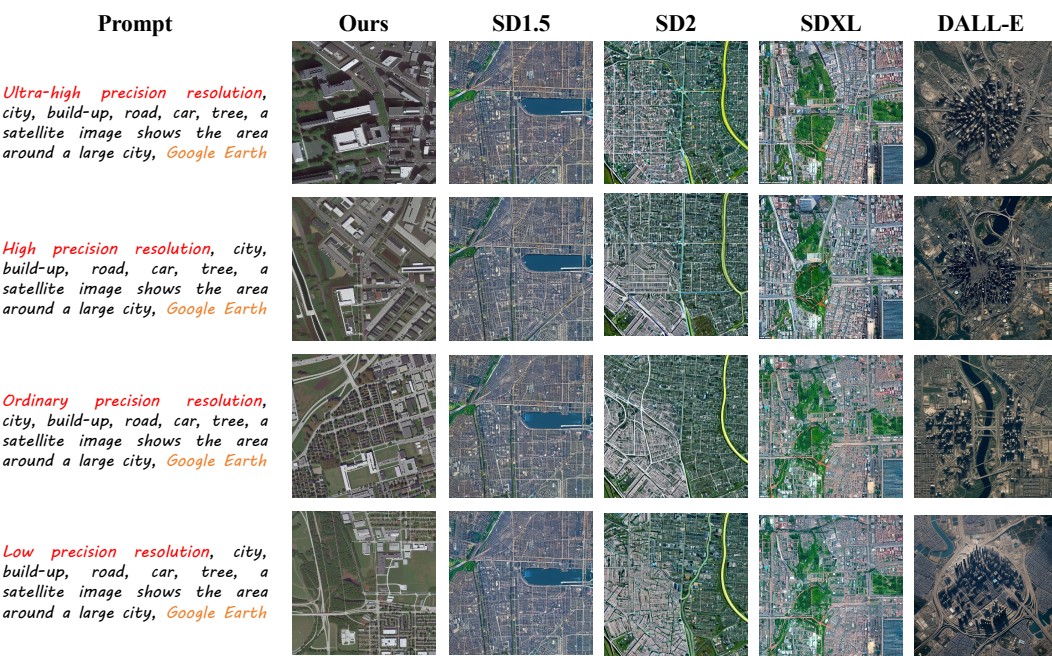

| Prompt | Ours | SD1.5 | SD2 | SDXL | DALL-E |

*Ultra-high precision resolution, city, build-up, road, car, tree, a satellite image shows the area around a large city, Google Earth*

*High precision resolution, city, build-up, road, car, tree, a satellite image shows the area around a large city, Google Earth*

*Ordinary precision resolution, city, build-up, road, car, tree, a satellite image shows the area around a large city, Google Earth*

*Low precision resolution, city, build-up, road, car, tree, a satellite image shows the area around a large city, Google Earth*

Figure 7: Visualization results of different methods in multi-GSD RS image generation. Our results show obvious variations of GSD according to the given text prompts.

**Qualitative comparison.** We generate multi-scene RS images using text prompts across various generative models, as depicted in Fig. 6. The results affirm our model's ability to accurately interpret and render diverse weather conditions. For example, the *"snow"* prompt yields images with detailed urban structures under snow cover in our results. In contrast, the results from the other methods

seem hardly adequate for snow scenarios. Similarly, our model can faithfully generate images with fog and night scenes, but other methods are hard to generate weather-consistent results. Therefore, the MMM-RS dataset not only enhances the quality and usability of generated RS images but also establishes a new standard for depicting weather and environmental conditions in synthetic images, reinforcing the purpose and significance of developing the MMM-RS dataset.

Fig. 7 shows the visualization results of different methods in multi-GSD RS image generation, we can observe that our results demonstrate clear GSD variations when generating RS images based on specific text prompts, highlighting our model's ability to adapt GSD settings effectively compared to other models. This adaptability allows our model to produce images ranging from ultra-high precision resolution to low precision resolution. For instance, the ultra-high precision resolution images reveal meticulous details such as building outlines and individual road lanes, which are distinctly visible. Conversely, the low precision resolution images, while less detailed, still maintain a level of clarity and contextual relevance suitable for broader landscape interpretations. These results prove that the MMM-RS dataset enables the model to perceive the various GSD through designated text prompts.

## 4.2 Cross-modal Generation based on ControlNet

In the above part, we conduct experiments to prove the validity of MMM-RS dataset in generating diverse RGB RS images. However, the multi-modal part remains to be further investigated. In this part, we aim to perform more interesting cross-modal generation experiments to verify the plausibility and validity of multi-modal data rather than simply fine-tuning the Stable Diffusion.

**Experiment settings.** We select the ControlNet [39] as the base model of cross-modal generation, which is a neural network architecture that can improve large-scale pretrained text-to-image diffusion models with input task-specific prior conditions. Concretely, we also use the pretrained Stable Diffusion-V1.5 model [23] as the backbone of the ControlNet, and the batch size is set to 4. We use a learning rate of 0.00005 and employ the Adam optimizer to train the models. For training the cross-modal generative models between RGB modality and SAR modality, we use approximately 290,000 RGB-SAR pairs in our MMM-RS dataset to train this model with 80,000 iterations. For training the cross-modal generative models between RGB modality and NIR modality, we use only 7,000 RGB-NIR pairs to train this model with 20,000 iterations.

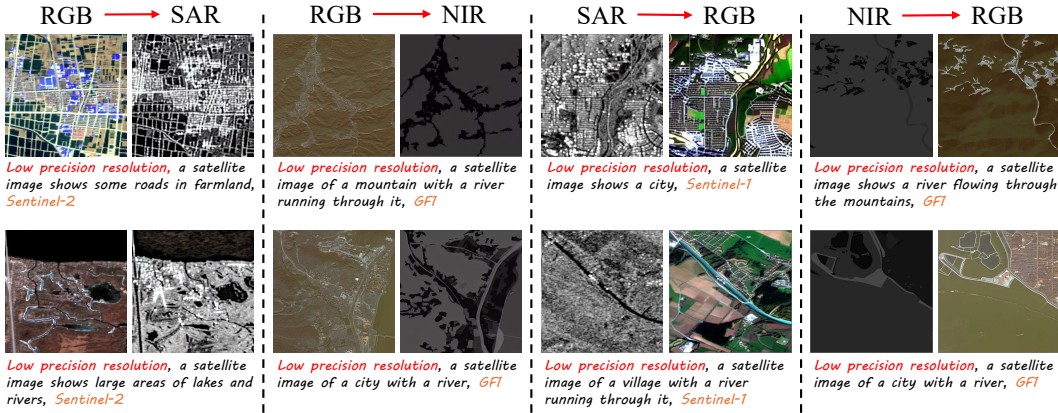

Figure 8: Visualization results of four different cross-modal generation tasks.

**Results of cross-model generation.** We conduct four different cross-model generation task: RGB → SAR, RGB → NIR, SAR → RGB, and NIR → RGB. Fig. 8 showcases the visualization results of the above four cross-model generation, we can observe that the generated SAR and NIR images from RGB → SAR and RGB → NIR can correctly depict the structural information of the input RGB images. For the SAR → RGB and NIR → RGB, the generated RGB images not only maintain the structural information of the input image but also exhibit rich textural details. The above results prove that our MMM-RS dataset can be effectively used for cross-modal generation tasks in RS scenarios.

# 5  Conclusion

In this paper, we propose a Multi-modal, Multi-GSD, Multi-scene Remote Sensing (MMM-RS) dataset and benchmark for text-to-image generation in diverse RS scenarios. MMM-RS is inspired by the investigation that there is no publicly available RS dataset that contains both multi-modal RS images and information-rich text descriptions for diverse and comprehensive RS image generation. Through the collection and standardization of nine publicly available RS datasets, we created a unified dataset comprising approximately 2.1 million well-crafted text-image pairs. With extensive experiments, we demonstrated the effectiveness of our dataset in generating multi-modal, multi-GSD, and multi-scene RS images.

# 6  Acknowledgement

This work was supported by the National Natural Science Foundation of China (Grants Nos. 62476133, 62372238, 62302219), the Natural Science Foundation of Shandong Province (Grant No. ZR2022LZH003), and the Natural Science Foundation of Jiangsu Province (Grant No. BK20220948).

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
