# Supplementary Materials for *MMM-RS: A Multi-modal, Multi-GSD, Multi-scene Remote Sensing Dataset and Benchmark for Text-to-Image Generation*

**Jialin Luo**[1,*], **Yuanzhi Wang**[1,*], **Ziqi Gu**[1], **Yide Qiu**[1], **Shuaizhen Yao**[1], **Fuyun Wang**[1], **Chunyan Xu**[1], **Wenhua Zhang**[1], **Dan Wang**[2], **Zhen Cui**[1,†]

1. PCA Lab, Key Lab of Intelligent Perception and Systems for High-Dimensional Information of Ministry of Education, School of Computer Science and Engineering, Nanjing University of Science and Technology, Nanjing, China.
2. Beijing Institute of Spacecraft System Engineering, Beijing, China.

## 1 Dataset Format and Composition

### 1.1 Annotation Format

The annotations of our proposed MMM-RS dataset are saved with the ***JSON*** format, and an example of annotation is shown in Fig. 1. Especially, in addition to our well-designed information-rich prompt, we maintain the labels from original dataset such as *"category"*, *"spatial_resolution"* (i.e., ground sample distance (GSD)), *"cloud_cover"* (i.e., levels of cloud cover), which could provide the user with more information for potential downstream tasks (e.g., classification and recognition).

```
{
"category": "educational_institution",
"spatial_resolution": 0.7993864844466622,
"cloud_cover": 0,
"timestamp": "2017-02-18T15:54:55Z",
"country_code": "USA",
"modality": "RGB",
"dataset": "FMOW",
"satellite": "WorldView",
"prompt": "Satellite imagery, High precision resolution, educational_institution, fog,  a satellite image of a campus with buildings and roads,  WorldView, FMOW",
"image_name": "educational_institution_660_7_msrgb_sr_fog.jpg",
"type": "Satellite imagery"
},
```

Figure 1: An example of annotation in our MMM-RS dataset.

### 1.2 Dataset Composition

Our MMM-RS dataset consists of nine datasets: MRSSC2.0 [3], Inria [4], NaSC-TG2 [9], GID [6], WHU-OPT-SAR [2], HRSC2016 [7], TGRS-HRRSD [8], SEN1-2 [5], and fMoW [1]. Below is a brief introduction of each dataset:

**MRSSC2.0** [3]: This dataset is constructed based on high-quality earth observation data obtained by the TianGong-2 Wideband Imaging Spectrometer and Interferometric Imaging Radar Altimeter. It is a cross-domain remote sensing scene classification dataset featuring four modes: Visible Near-Infrared (VIS), Short Wavelength Infrared (SWI), Thermal Infrared (INF), and Synthetic Aperture Radar (SAR). However, the images in different modalities are not aligned. This paper only uses the RGB

---

*Co-first Authors, Equal Contribution: Jialin Luo, Yuanzhi Wang
†Corresponding Author: Zhen Cui

38th Conference on Neural Information Processing Systems (NeurIPS 2024) Track on Datasets and Benchmarks.

images from this dataset, which include seven classic scenes such as mountains, rivers, and lakes. It contains a total of 6155 images, each with an original pixel resolution of $256 \times 256$ and an original GSD of 100 m/pixel. After standardized processing, the actual GSD is 50 m/pixel and the number of images is 6155.

**Inria** [4]: This aerial dataset covers an area of 810 square kilometers, containing a total of 360 large images of $5000 \times 5000$ pixels with a GSD of 0.3 m/pixel. These images cover different urban residential areas, ranging from densely populated regions to mountainous towns. After standardized processing, the actual GSD is 0.3 m/pixel and the number of images is 2872.

**NaSC-TG2** [9]: This dataset is a novel remote sensing natural scene classification benchmark dataset based on TianGong-2 remote sensing images. It aims to expand and enrich annotated data to promote remote sensing classification algorithms, especially natural scene classification. The dataset contains 20,000 RGB images, evenly divided into 10 scene categories. Each image is $128 \times 128$ pixels, with a GSD of 100 m/pixel. After standardized processing, the actual GSD is 25 m/pixel and the number of images is 20,000.

**GID** [6]: This dataset is a large-scale high-resolution remote sensing image land cover dataset based on Gaofen-2 (GF-2) satellite data. It consists of 150 images with $6800 \times 7200$ pixels, including five land cover categories: buildings, farmland, forests, grasslands, and water bodies. After standardized processing, the actual GSD is 0.8 m/pixel and the number of images is 31,500.

**WHU-OPT-SAR** [2]: This dataset utilizes optical images from the Gaofen-1 (GF-1) satellite and SAR images from the Gaofen-3 (GF-3) satellite of the same region. It consists of RGB, Near-Infrared (NIR) optical images, and corresponding SAR images, covering an area of 51448.56 km$^2$ with a GSD of 5 m/pixel. It includes 100 pairs of images with $5556 \times 3704$ pixels (three modalities), covering six land cover categories. After standardized processing, the actual GSD is 5 m/pixel and the number of images is 21,000.

**HRSC2016** [7]: This dataset is a ship object detection dataset with a GSD of 0.4-2 m/pixel. After standardized processing, the number of images is 1680.

**TGRS-HRRSD** [8]: This dataset includes 21,761 images, with a total of 55,740 object instances and corresponding labels (such as ships, bridges, tennis courts, etc.), sourced from Google Earth and Baidu Maps. The actual GSD ranges from 0.15 to 1.2 m/pixel.After standardized processing, the number of images is 21,761.

**SEN1-2** [5]: This dataset consists of RGB images and SAR images from Sentinel-1 and Sentinel-2, with aligned scenes, comprising approximately 560,000 RGB and SAR images. Each image is $256 \times 256$ pixels, with a GSD of 10 m/pixel. After standardized processing, the actual GSD is 5 m/pixel and the number of images is 560,000.

**fMoW** [1]: This dataset is well-annotated, containing information such as timestamp, cloud coverage, GSD, categories, etc. The GSD ranges from 0.5 m to 2 m/pixel. This paper manually screens about 930,000 original images, which are cropped and processed into 1.43 million images after standardized processing.

## 2   Visualization of Generated Images via Different Satellite Prompts

Considering the heterogeneous imaging styles of cameras on different satellites, we inject the satellite names as part of the text prompts, thereby allowing the models to generate images with various satellite imaging styles. Fig. 2 shows the visualization results of generated images via different satellite prompts in city scene, we can observe two conclusions: **1)** the generated images with different satellite prompts show diverse imaging styles. **2)** Comparing with the reference images from our dataset, the generated images are almost consistent with the reference images in the imaging style of different satellites, which proves that the satellite prompts are feasible and effective.

## 3   Generalization of Prompts

To validate the generalization performance of our prompts, we have also provided additional visualization results in Fig. 3, where we alter the sequence of contents in the prompts and examine the minor variations in the results.

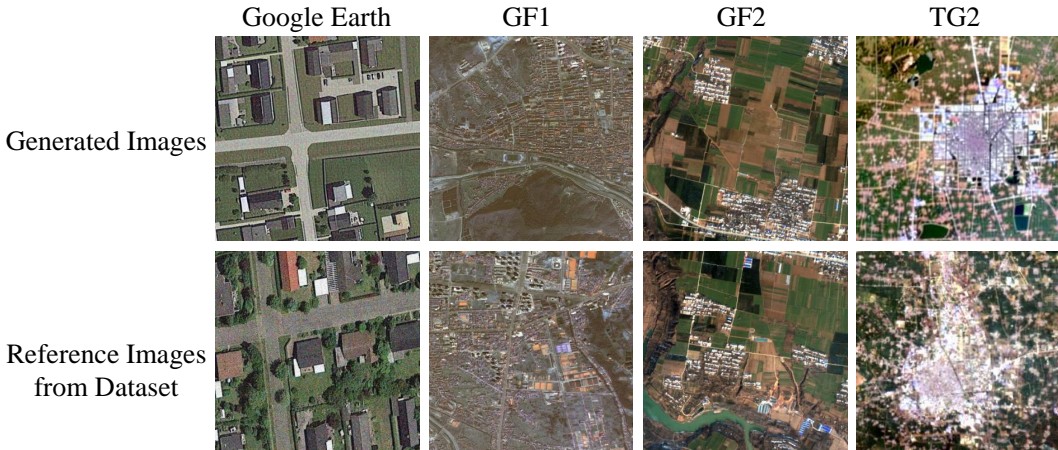

Figure 2: Visualization results of generated images (city scene) via different satellite prompts.

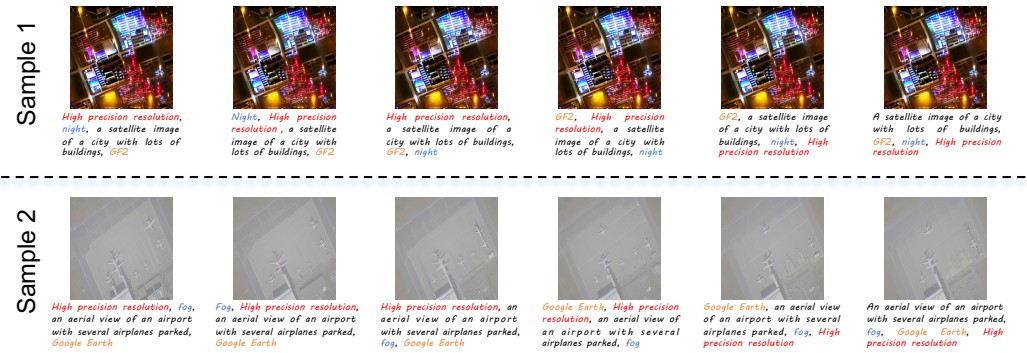

Figure 3: Visualization results of the different prompts with the same text content.

## 4 Additional Quantitative Comparisons

In this section, we conduct additional experiments and provide additional quantitative comparisons. Tab. 1 reports the quantitative results for stable diffusion v1.5 (SD1.5), stable diffusion v2.0 (SD2.0), and stable diffusion XL (SDXL), we can observe that our proposed MMM-RS dataset can significantly enhance the generative capabilities of existing models for remote sensing scenarios.

Table 1: Comparison of FID/IS scores for different models.

| Model | SD1.5 | SD2.0 | SDXL |
|---|---|---|---|
| Base | 172.78/6.65 | 175.68/6.31 | 168.34/6.89 |
| Fine-tune | **92.33/7.21** | **134.21/6.87** | **129.62/7.26** |

## 5 Additional Visualization Results of Cross-modal Generation

We provide the additional visualization results of cross-modal generation in Fig. 4, which shows the results in the high precision resolution scene.