# OpenReview forum: "MMM-RS: A Multi-modal, Multi-GSD, Multi-scene Remote Sensing  Dataset and Benchmark for Text-to-Image Generation"
_NeurIPS.cc/2024/Datasets_and_Benchmarks_Track — NeurIPS 2024 Track Datasets and Benchmarks Poster_

### Official Review · Reviewer_xUzE · 2024-07-18
**An Interesting Multi-modal RS Text-to-Image Generation Dataset**

**Rating:** 7
**Confidence:** 4
**Correctness:** Yes
**Clarity:** Yes

**Review:**

Please see Strengths and Limitations

**Strengths:**

1. The proposed dataset includes various image modalities such as SAR, NIR, and RGB, enabling cross-modal generation.

2. The dataset focuses on crucial features of remote sensing data, including GSD, weather, and capturing devices, filling a significant gap in current text-image datasets for remote sensing. This enhances the diversity and accuracy of generated remote sensing images.

3. The experiments conducted are comprehensive, covering qualitative and quantitative analyses, and showcases the dataset's advantages from various perspectives.

4. The paper is well-structured, with a clear and logical flow, and is highly readable.

**Additional Feedback:**

Is there a specific rationale for the division of GSD levels, and could a finer-grained division be employed?

**Documentation:**

The dataset link is provided

**Limitations:**

1. There is an imbalance in the image modalities, with RGB images being much more prevalent than others. It can be considered that use image translation and manual selection methods to expand other modality images.

2. Some datasets used in the paper already contain text prompts, which could potentially be used to make the BLIP-generated text prompts more accurate.

3. The well-structured prompts used in the paper raise concerns about generalization. I believe it would be beneficial to add experiments to demonstrate the fine-tuned model's generalization ability, by using more generic prompts.

4. The experiments could be more convincing if they included comparisons with models fine-tuned on other multimodal remote sensing datasets, rather than using the original models. This would better highlight the advantages of the proposed dataset and strengthen the paper's arguments.

**Opportunities For Improvement:**

Please address the issues in the limitations.

**Relation To Prior Work:**

Yes

**Summary And Contributions:**

This paper introduces the Multi-modal, Multi-GSD, Multi-scene Remote Sensing (MMM-RS) dataset for remote sensing image generation tasks. This dataset includes not only typical text-image pairs but also various paired remote sensing image modalities. Additionally, this paper extracts key remote sensing image features, including GSD, weather and satellite to use as parts of prompts. Furthermore, the dataset is expanded using text generation and image synthesis methods. Experimental results show that models fine-tuned on this dataset perform better with corresponding prompts. The experiments also demonstrate the capability of cross-modal generation.

---

> ### Author Rebuttal · Authors · 2024-08-17
>
> ## Thanks for recognizing our innovation and promising results. Below, we respond to each Question (Q) with an Answer (A).
>
> **Q1:** There is an imbalance in the image modalities, with RGB images being much more prevalent than others. It can be considered that use image translation and manual selection methods to expand other modality images.
>
> **A1:** Thanks for your suggestion. As existing publicly available aligned multi-modal datasets are very scarce, in the current version, we have tried to synthesize some multi-modal data by the advanced generative methods based on RGB images. Also, we will consider the methods you said (image translation and manual selection) to extend the other modalities in the next dataset update. Thanks again.
>
> **Q2:** Some datasets used in the paper already contain text prompts, which could potentially be used to make the BLIP-generated text prompts more accurate.
>
> **A2:** Thanks for your suggestion. In fact, during the manual cleaning phase, we use the original information (e.g., input images, prompts, labels) from the datasets to manually correct the inaccurate prompts predicted by the BLIP model.
>
> **Q3:** The well-structured prompts used in the paper raise concerns about generalization. I believe it would be beneficial to add experiments to demonstrate the fine-tuned model's generalization ability, by using more generic prompts.
>
> **A3:** Thanks for your suggestion. Our prompt format follows the mainstream format of existing text-to-image models, please see Stable Diffusion web UI: [Stable Diffusion Web UI](https://github.com/AUTOMATIC1111/stable-diffusion-webui). Thus, these prompts are inherently generic. Furthermore, to address your concerns about the model's generalization capabilities, we conduct additional visualization experiments. Specifically, we alter the sequence of the contents within the prompts and show the corresponding visual results. Fig. 1 in our uploaded PDF file (in the rebuttal stage) shows some of the visual results, we can observe that the generated results from the different prompts with the same text content are almost identical (with only minor variations in details). These results demonstrate that the content generated by the model is not constrained by the format of the prompts in the training dataset, indicating a high generalization ability.
>
> **Q4:** The experiments could be more convincing if they included comparisons with models fine-tuned on other multimodal remote sensing datasets, rather than using the original models. This would better highlight the advantages of the proposed dataset and strengthen the paper's arguments.
>
> **A4:** Thanks for your suggestion. It needs to be clarified that, to our knowledge, there are currently no other publicly available multi-modal RS text-to-image datasets for model fine-tuning. The motivation behind this paper is to fill this gap in the dataset availability in this field. We hope that the proposed dataset will drive the development of this field and encourage more researchers to study and enhance the capabilities of multi-modal RS text-to-image generation. Thank you once again for your valuable and insightful suggestions.
>
> **Q5:** Is there a specific rationale for the division of GSD levels, and could a finer-grained division be employed?
>
> **A5:** The division of GSD levels in our dataset is primarily based on the concentrated distribution of GSD values within the images of these datasets. Furthermore, different GSD levels exhibit significant visual differences, which makes it easier for the training of generative models to learn the distinctions between various GSD levels. In addition, more finer-grained division of GSD levels could be extended from our current version.

---

> > ### Comment · Reviewer_xUzE · 2024-08-26
> > **Official comments by Reviewer xUzE**
> >
> > The author's response has addressed most of my concerns, particularly regarding generalization and comparison experiments. Although the imbalance in data modalities will be addressed in the next update, this version already demonstrates a substantial amount of work. Therefore, I will maintain my original score.

---

> > > ### Author Response · Authors · 2024-08-26
> > > **Thank you for reviewing our response and recognizing our work!**
> > >
> > > Thank you for reviewing our response and recognizing our work! We are pleased that our response addressed your concerns!
> > >
> > > For the multi-modal RS dataset, the imbalance of modalities is an inherent limitation of the RS field (i.e., aligned multi-modal data collection is very difficult), we will continue to follow up and update the multi-modal part in our MMM-RS dataset.
> > >
> > > Thank you once again for your valuable comments and suggestions!

---

### Official Review · Reviewer_7rKP · 2024-07-21
**Good work although I have some concerns.**

**Rating:** 6
**Confidence:** 5
**Correctness:** All good.
**Clarity:** Yes, it is.

**Review:**

Quality: the work is well written and easy to follow. I didn't find any grammatical errors.
Originality: The authors are working on a problem other researchers are also working on with a small twist towards image generation, which makes it slightly original.
Pros: Large text2image dataset that authors will make available.
Cons: The authors compare their model against four different baselines, being one of them the model they use to fine-tune. In the paper, they mention that they compare using (FID, IS) the generated images to real images. I am concerned authors may have used the same type of images they used to train their model. Moreover, the prompts they use were tailored to their model, the other models may have been generating better images with other prompts.
Not sure this may be the right place, but the authors release their dataset under a CC BY 4.0 license. INRIA dataset uses data that may be in conflict with this license, please check the European imagery. fMOW does not allow commercial use if I am not wrong.
It is not full clear to me how the authors did the hand-crafted cleanse.
The dataset is not fully accesible (e.g. prompts).

**Strengths:**

Large dataset publicly available. Well written paper. Clear and simple experimental setup.

**Additional Feedback:**

Please use a more friendly way of sharing your dataset, and provide the whole dataset including prompts. Do not host datasets that belong to others unless they provide their consent.

**Documentation:**

The authors do not provide the prompts in their dataset, one of the links is not working.

**Ethics:**

Just double check what I mentioned earlier about the license. Additionally, I must alert the authors that they are hosting datasets on a platform (I may be wrong) without the consent of the original authors.

**Limitations:**

Please check the license of each of the datasets you are using to build yours. If you release the dataset under a license that is in conflict with these you may have a negative impact.

**Opportunities For Improvement:**

I'd suggest authors to add to their paper some qualitative experiments by showing the images and the prompt to anonymous users (with some RS background) and let them ranking the generated images.

**Relation To Prior Work:**

It does seem to include all the relevant previous work.

**Summary And Contributions:**

The authors, using nine publicly available datasets, present a new large dataset that includes RGB, NIR and SAR images together with rich-text information. They alter the images using available algorithms in the literature and use BLIP-2 to extract text information out of each image. They later fine-tune using LoRA a VLM (Stable Difussion v1.5) with their dataset and conduct a comparison against SD1.5, SD2.0, SDXL and Dall-E 3 showing their dataset can yield better performance (FID, IS).

---

> ### Author Rebuttal · Authors · 2024-08-17
>
> ## Thanks for recognizing our innovation and promising results. Below, we respond to each Question (Q) with an Answer (A).
>
> **Q1:** The authors compare their model against four different baselines, being one of them the model they use to fine-tune. In the paper, they mention that they compare using (FID, IS) the generated images to real images. I am concerned authors may have used the same type of images they used to train their model.
>
> **A1:** Thanks for your comment. We make the clarification as follows:
>
> i) The question you raised pertains to Table 2 in our manuscript. This table aims to validate the effectiveness of our dataset for training generative models, by incorporating an incremental module for fine-tuning into each base network. The comparisons highlight the necessity of our specialized remote sensing imagery dataset.
>
> ii) In the comparisons of Table 2, we strictly follow the training paradigm and evaluation protocol (including FID and IS) as used in the field of image generation. For the two metrics, FID (Fréchet Inception Distance) calculates the Fréchet distance between two distributions to evaluate the differences between the distributions of generated data and real data; Inception Score (IS) is to evaluate the quality of images, and its scoring calculation does not require the comparison data. Please note that the used evaluation ways are standard in the current study of image generation.
>
> For your concern, we will re-clarify these points in the revision.
>
> **Q2:** The prompts they use were tailored to their model, the other models may have been generating better images with other prompts.
>
> **A2:** Thanks for your comment. Our prompt format follows the mainstream format of existing text-to-image models, please see Stable Diffusion web UI: [Stable Diffusion Web UI](https://github.com/AUTOMATIC1111/stable-diffusion-webui). In other word, these prompts are not tailored to our model. Additionally, to further address your concern, we conduct additional visualization experiments. Specifically, we alter the sequence of the contents within the prompts and show the corresponding visual results. Fig. 1 of our uploaded PDF file (in the rebuttal stage) shows some of the visualization results, we can observe that the generated results from the different prompts with the same text content are almost identical (with only minor variations in details). These results demonstrate that the content generated by the model is not constrained by the format of the prompts.
>
> **Q3:** Please check the license of each of the datasets you are using to build yours. If you release the dataset under a license that is in conflict with these you may have a negative impact.
>
> **A3:** Thanks for your suggestions and carefully reviewing our work. We have carefully checked the licenses of all datasets we used and have determined a final license: CC BY-NC-SA 4.0 (i.e., Attribution-NonCommercial-ShareAlike 4.0 International), which meets the restrictive requirements of all datasets. We appreciate your timely corrections regarding our license and your help in avoiding any negative impacts that could arise from potential conflicts with existing datasets.
>
> **Q4:** How the authors did the hand-crafted cleanse.
>
> **A4:** Thanks for your question. The hand-crafted cleaning process primarily involves checking, modifying, and enhancing the initial prompts generated by BLIP to ensure the inclusion of diverse information, such as objects, scenes, weather conditions, etc. For the final prompt of each image, we arrange for three persons to do this cleanse and summarize their results.
>
> **Q5:** The dataset is not fully accessible (e.g. prompts).
>
> **A5:** For your question, we have re-uploaded the dataset to other link: OneDrive (link: [OneDrive](https://1drv.ms/f/c/ac80b86748dd94c5/EoRUSpXBc6VAqgl3rd9Qz_IBBxrEhObC3b-JIMUXjmvkyw)), and we have tested that both this link and the original link are valid. The metadata, including prompts, is contained within JSON files in each subfolder.
>
> **Q6:** I'd suggest authors to add to their paper some qualitative experiments by showing the images and the prompt to anonymous users (with some RS background) and let them ranking the generated images.
>
> **A6:** Thanks for your valuable suggestion. We conduct a user study to compare the RS images generated by different methods. Due to the time limitation of the rebuttal stage, we create a survey that includes 60 prompt-image pairs (12 pairs per method), and participants could select whether each text-image pair is aligned. Based on the results, we can calculate the human satisfaction rate (HSR) of the generated images for each method using the following equation: $\text{HSR} = \left( \frac{N_{\text{approved}}}{N_{\text{total}}} \right) \times 100 $%. We also provide some cases of this survey in Fig. 3 of our uploaded PDF file. The survey is distributed to 33 researchers with the RS background during the limited rebuttal time. The results are listed in the following table, we can observe that the model fine-tuned with our dataset achieved a higher HSR. All details of the user study will be included in the revision.
>
> | Metric | SD1.5  | SD2.0  | SDXL   | DALL-E 3 | Ours |
> |--------|--------|----|-----|------|---|
> | HSR    | 82.76% | 84.55% | 85.12% | 62.33%  | 91.25% |
>
> **Q7:** Additionally, I must alert the authors that they are hosting datasets on a platform (I may be wrong) without the consent of the original authors.
>
> **A7:** Thanks for your reminding. We carefully check the licenses of all datasets we used and confirm that non-commercial use, modification, and sharing are permitted. At the current review stage, all materials uploaded to the provided links have been reprocessed and re-annotated by our team. In the forthcoming public release, we will only provide all processing codes, prompt annotations and our private data. According to the release, users could reproduce the version of our dataset after they download the involved datasets.

---

> > ### Comment · Reviewer_7rKP · 2024-08-18
> > **Recommendation for Acceptance**
> >
> > I am pleased with the authors' response. They have thoroughly addressed the concerns (license and redistribution mostly) and carried out the recommended additional experiments. Therefore, I recommend accepting the paper.

---

> > > ### Author Response · Authors · 2024-08-18
> > > **Thank you for recognizing our work!**
> > >
> > > Thank you for reviewing our response and recognizing our work! We are pleased that our response addressed your concerns.

---

### Official Review · Reviewer_8np9 · 2024-07-25
**Comprehensive dataset; solid analysis; improvement needed**

**Rating:** 6
**Confidence:** 4
**Clarity:** Besides the issue mentioned above, th…

**Review:**

See the strengths and weaknesses below.

**Strengths:**

1. The proposed dataset includes various image modalities and image-text pairs, filling a gap in text-image RS datasets.

2. The experiments results are solid, including qualitative and quantitative analyses, from aspects of text-to-image generation under different weather conditions and GSD levels, and cross-modal generation.

3. The paper is well-written, clear, and logically presented.

**Additional Feedback:**

1. Real surface reflectance in remote sensing images, particularly under clear weather conditions, is crucial for accurate image interpretation. What is the benefit of generating RS images under a variety of weather conditions?

**Correctness:**

Besides the raised issue in ‘Opportunities For Improvement’, the dataset is constructed in a sound way.

**Documentation:**

Yes, there are links to the github repo and dataset.

**Ethics:**

No ethical concerns.

**Limitations:**

No limitations are discussed.

**Opportunities For Improvement:**

1. The multi-scene dataset is synthesized using classic atmospheric scattering-based degradation model, TPSeNCE, CycleGAN for fog, low-light, and snow scenes, respectively. The effect of using these synthesis methods to generate multi-scene data should be analyzed. Moreover, why not consider using real images captured under these weather conditions to form the dataset?

2. More common weather condition is not considered, such as cloud cover.

3. Typo in L247-L250:’cross-model’ should be ‘cross-modal’.

4. Section 4.2: It should be specified whether the GSD level in cross-modal generation tasks should be restricted to 'Low precision resolution'.

**Relation To Prior Work:**

The review of related work is comprehensive. Also, the cutting-edge method, such as ControlNet is evaluated in the experiment.

**Summary And Contributions:**

This paper proposes a Multi-modal, Multi-GSD, Multi-scene Remote Sensing (MMM-RS) dataset for text-to-image generation. This dataset contains approximately 2.1 million text-image pairs. This dataset helps diffusion models generate diverse RS images across various modalities, scenes, weather conditions, and GSD.

---

> ### Author Rebuttal · Authors · 2024-08-17
>
> ## Thanks for recognizing our innovation and promising results. Below, we respond to each Question (Q) with an Answer (A).
>
> **Q1:** The multi-scene dataset is synthesized using classic atmospheric scattering-based degradation model, TPSeNCE, CycleGAN for fog, low-light, and snow scenes, respectively. The effect of using these synthesis methods to generate multi-scene data should be analyzed. Moreover, why not consider using real images captured under these weather conditions to form the dataset?
>
> **A1:** Thanks for your comment. We make the clarification as follows:
>
> i) As multi-weather RS samples are rare for publicly available and difficult to collect, in particular, collecting *paired multi-weather data* of the same scene is extremely difficult. To ensure the comprehensiveness and completeness of our proposed dataset, as well as to enable the well-trained model to generate samples under different weather conditions, we have to choose advanced methods to synthesize some multi-weather data, thereby enhancing the dataset's diversity. Furthermore, *we manually check the synthesized data to minimize noise as much as possible*.
>
> ii) The synthesizing models (atmospheric scattering-based degradation model, TPSeNCE, CycleGAN) for weather data are widely recognized and effective. To further validate the effectiveness of these synthesized multi-weather data, we conduct an additional set of ablation experiments, where the model is trained with only 1/5 of the synthesized weather data, and quantitatively compared with the model trained with the complete dataset. The following table lists the quantitative FID results of samples generated under three weather scenarios, we can observe a considerable decrease in performance when training with less data compared to using the full dataset, which consistently validates the effectiveness of these synthesized weather data.
>
> | Dataset Size | Fog    | Snow   | Night  |
> |-|-|-|-|
> | 1/5| 196.32 | 214.22 | 210.63 |
> | **All**|**153.89**|**172.41**|**168.72**|
>
> iii) In the future, additional real-world multi-weather remote sensing samples may be seamlessly integrated into the dataset without necessitating modifications to the existing framework of the dataset.
>
> **Q2:** More common weather condition is not considered, such as cloud cover.
>
> **A2:** Thanks for your suggestion. In fact, our dataset contains a proportion of real cloud cover samples, although this was not emphasized in the manuscript. The reason is that, in practice, constructing well-paired multi-weather samples (with/without clouds) of the same scene is difficult, as the current methods for cloud generation or removal we have tested are not adequately effective. We are actively striving to collect additional weather samples, including those with cloud cover, to further enrich the diversity of the dataset.
>
> **Q3:** Typo in L247-L250: 'cross-model' should be 'cross-modal'.
>
> **A3:** Thanks for careful reading of our paper. We will correct this typo and polish our manuscript carefully in the revision.
>
> **Q4:** Section 4.2: It should be specified whether the GSD level in cross-modal generation tasks should be restricted to 'Low precision resolution'.
>
> **A4:** For the question you said, it is related to the visualization results in Fig. 8 in our manuscript. In this figure, we only exhibited the result of `low precision resolution' mainly due to space limitations. Our method is not limited to this case. For your concerns, we provide some additional visualization results with high precision resolution in Fig. 2 of our uploaded PDF file in the rebuttal stage.
>
> **Q5:** Real surface reflectance in remote sensing images, particularly under clear weather conditions, is crucial for accurate image interpretation. What is the benefit of generating RS images under a variety of weather conditions?
>
> **A5:** Thanks for your comment. While clear weather conditions are crucial for remote sensing, a greater diversity of weather conditions (such as fog, snow, etc) is essential for real-world applications. In practical applications, remote sensing images under various weather conditions often need to be collected for training models (e.g., scene understanding, object detection/recognition, etc). By generating images across a range of weather conditions, we can enhance the adaptability of trained models to complex scenes. For instance, an object detection model typically performs poorly in fog conditions if it has only been trained on data from clear weather.

---

> > ### Comment · Reviewer_8np9 · 2024-08-31
> > **Post-rebuttal discussion**
> >
> > Regarding the 'multi-weather' issue, frequent remote sensing observations can provide multi-weather RS samples, which may be considered in future research. Overall, the authors have addressed most of my questions, and therefore, I am considering maintaining the rating.

---

> > > ### Author Response · Authors · 2024-08-31
> > > **Thank you for recognizing our work!**
> > >
> > > Thank you for reviewing our response! We are pleased that our response addressed your concerns!
> > > For the "mulit-weather" issue, we will continue to follow up and update the dataset for more mulitple weather scenarios.
> > > Once again, thank you very much for your conmments and suggestions.

---

### Official Review · Reviewer_DrKY · 2024-07-29

**Rating:** 5
**Confidence:** 4
**Correctness:** 1. The multi-scene Remote Sensing Ima…
**Clarity:** There is ample room for enhancing the…

**Review:**

The motivation is commendable, yet there is ample room for enhancing the coherence and precision of the writing. Please consider the "Opportunities For Improvement."

**Strengths:**

The motivation is commendable.

**Additional Feedback:**

N/A

**Documentation:**

1. The dataset is selected to be made public at the review stage, it is suggested to detail the dataset information on the linked website.
2. The dataset information needs to be elaborated further in the main paper.

**Limitations:**

1. The authors assert "Information-rich Text Prompt Generation," yet it appears to be text with a fixed format.

**Opportunities For Improvement:**

1. Since the dataset is selected to be made public at the review stage, it is suggested to detail the dataset information on the linked website.

2. It is advisable to elaborate on Table 1. For instance, based on the main paper descriptions, it would be beneficial to specify if the text descriptions are information-rich and clarify the meaning of "modality" in the context of multi-modal data. These are crucial details that should be included in Table 1.

3. The content in Section 2, Background, should align with Table 1 for improved clarity and coherence.

4. In Section 3.1, detailed descriptions of 9 datasets within the newly established MMM-RS dataset are essential, or the subsequent statistics of MMM-RS remain unclear.

**Relation To Prior Work:**

This work is innovative.

**Summary And Contributions:**

This paper introduces the MMM-RS dataset and benchmark for diverse remote sensing image generation, tackling the issue of sparse comprehensive datasets. The MMM-RS dataset is curated by amalgamating public RS datasets, generating textual descriptions, and diversifying images across various GSDs and environments.

---

> ### Author Rebuttal · Authors · 2024-08-17
>
> ## Thanks for recognizing our innovation and promising results. Below, we respond to each Question (Q) with an Answer (A).
>
> **Q1:** It is suggested to detail the dataset information on the linked website.
>
> **A1:** Thanks for your suggestion. According to your valuable suggestion, we add more details about the dataset in the provided link. This includes an organized structure of the dataset, information on modalities, visual diagrams, and several generated results, which allows readers to gain a more comprehensive understanding of our dataset through the website.
>
> **Q2:** It is advisable to elaborate on Table 1. It would be beneficial to specify if the text descriptions are information-rich and clarify the meaning of "modality" in the context of multi-modal data.
>
> **A2:** Thanks for your suggestion. According to your valuable suggestion, we update Table 1 to specify whether the text descriptions are information-rich and to clarify the concrete modalities when referring to multi-modal data. This update is included in Tab. 1 (Blue fonts) of our uploaded PDF file in the rebuttal stage.
>
> **Q3:** The content in Section 2, Background, should align with Table 1 for improved clarity and coherence.
>
> **A3:** Thanks for your suggestion. In the background part, our original purpose was to introduce several classic datasets used in various remote sensing tasks. According to your valuable suggestion, we will re-clarify the description of the background section in the revision to ensure it clearly aligns with the datasets presented in Table 1.
>
> **Q4:** In Section 3.1, detailed descriptions of 9 datasets within the newly established MMM-RS dataset are essential.
>
> **A4:** Thanks for your suggestion. Due to space limitations, the detailed descriptions of each dataset have been deferred to our submitted Supplementary Material (please see Section 1.2: Dataset Composition).
>
> **Q5:** The authors assert "Information-rich Text Prompt Generation," yet it appears to be text with a fixed format.
>
> **A5:** There might be some misunderstanding. We make the explanation in two points:
>
> i) The term \`\`Information-rich" refers to more descriptions (e.g., GSD levels, weather information, type of satellite, etc) of remote sensing domain for images, rather than fixed format.
>
> ii) The format of prompts here is constructed in analogy with those text-to-image models, such as Stable Diffusion.
>
> iii) To further address your concern, we new add an additional experiment to perturb the prompt sequence. The visual results are shown in Fig. 1 of our uploaded PDF file in the rebuttal stage. We can observe that the generated results from the disturbed prompts with the same text content are nearly identical (with only minor variations in details), demonstrating that Information-rich Text Prompt Generation is largely independent of any fixed format.
>
> **Q6:** The multi-scene Remote Sensing Images are generated by a model, which inevitably introduces noise. Could this noise impact the performance of a model trained on the benchmark with noisy data?
>
> **A6:** For your question, we make the clarification as follows:
>
> i) As multi-weather RS samples are rare for publicly available and difficult to collect, in particular, obtaining *paired multi-weather data* of the same scene is extremely difficult. To ensure the comprehensiveness and completeness of our proposed dataset, as well as to enable the well-trained model to generate samples under different weather conditions, we have to choose advanced methods to synthesize some multi-weather data, thereby enhancing the dataset's diversity. Furthermore, *we manually check the synthesized data to minimize noise as much as possible*.
>
> ii) The synthesizing models for weather data are widely recognized and effective. To further validate the effectiveness of these synthesized multi-weather data, we conduct an additional set of ablation experiments, where the model is trained with only 1/5 of the synthesized weather data, and quantitatively compared with the model trained with the complete dataset. The following table lists the quantitative FID results of samples generated under three weather scenarios, we can observe a considerable decrease in performance when training with less data compared to using the full dataset, which consistently validates the effectiveness of these synthesized weather data.
>
> | Dataset size | Fog| Snow| Night|
> |-|-|-|-|
> |1/5| 196.32| 214.22| 210.63|
> |All| **153.89**| **172.41** | **168.72** |
>
> iii) In the future, additional real-world multi-weather remote sensing samples may be seamlessly integrated into the dataset without necessitating modifications to the existing framework of the dataset.
>
> **Q7:** How were the results in Table 2 obtained? Are the results of other methods zero-shot outcomes? Do the results of "Ours" correspond to training on the proposed MMM-RS dataset? Despite some related descriptions preceding Table 2, the information remains ambiguous yet is of significant importance.
>
> **A7:** We apologize for any ambiguity that may have caused you confusion. Here, we respond to your concerns:
>
> i) The results in Table 2 are calculated on randomly generated 500 samples for each model separately (please also see Lines 204-206 of our paper).
>
> ii) The results for the comparison methods are also obtained in a zero-shot manner.
>
> iii) \`\`Ours'' refers to the model fine-tuned on our constructed MMM-RS dataset, we use the pretrained Stable Diffusion (SD) 1.5 model as the base model (please see Lines 187-189). Here we also provide the comparisons with other base models (SD2.0 and SDXL) as shown in the following table, which further demonstrates the effectiveness.
>
> | Models| SD1.5| SD2.0| SDXL|
> |-|-|-|-|
> || FID↓/IS↑| FID↓/IS↑| FID↓/IS↑|
> | Base model| 172.78/6.64| 175.68/6.31|168.34/6.89|
> | Fine-tuned  model| **92.33/7.21**|**134.21/6.87**|**129.62/7.26**|
>
> According to your comments, we make sure to re-clarify the descriptions and add more comparisons (above) in the revision.

---

> > ### Comment · Reviewer_DrKY · 2024-08-19
> > **Official comments by Reviewer DrKY**
> >
> > 1. The authors assert "Information-rich Text Prompt Generation," yet it appears to be text with a fixed format. That is what i am trying to say: the fixed-format text contradicts the concept of information-rich text; text with an open and flexible format tends to be more information-rich.
> > 2. In my view, model-generated images should contain noise, potentially impacting model training. This aspect should be discussed in a well-rounded paper.
> >
> > Additionally, the authors have addressed the concerns raised. I am considering raising the rating. If accepted, please meticulously revise the paper in accordance with the reviewers' suggestions.

---

> > > ### Author Response · Authors · 2024-08-19
> > > **Thank you for reviewing our response and increasing your rating!**
> > >
> > > We appreciate your thoughtful review of our response and are pleased to note the increase in your rating. We are pleased that our response addressed your concerns!
> > >
> > > Regarding the two comments you summarized from the first-round review, we understand the need for clarification in the revised manuscript, as previously addressed in our rebuttal. We assure you that all revisions and additional experiments in rebuttal will be incorporated into the accepted version.
> > >
> > > Thank you once again for your valuable comments and suggestions!

---

### Author Rebuttal · Authors · 2024-08-17

Dear Reviewers,

We would like to thank you for your valuable comments and suggestions. We have carefully made the corresponding response for each comment, which is as follows. **In particular, the additional experimental results are added in the PDF file.** We hope these responses could address your concerns.

Thank you so much!

---

### Decision · Program_Chairs · 2024-09-26

**Decision:**

Accept (Poster)

**Comment:**

This paper initially received three positive ratings and one negative rating. After reviewing the authors' responses, Reviewer #DrKY, who provided the negative rating, is inclined to raise their score. The paper is well-motivated, presents a dataset with diverse image modalities and image-text pairs, and makes the dataset publicly available. The experimental results are solid, and the paper is well-written. As a result, the Area Chair (AC) recommends to accept the paper.